# Long Term Outcomes of No-Touch Isolation Principles Applied in Pancreaticoduodenectomy for Treatment of Pancreatic Adenocarcinoma: A Multicenter Retrospective Study with Propensity Score Matching

**DOI:** 10.3390/jcm12020632

**Published:** 2023-01-12

**Authors:** Yu Mou, Yi Song, Jinheng Liu, Haiyu Song, Xubao Liu, Jiang Li, Nengwen Ke

**Affiliations:** 1Department of Pancreatic Surgery, West China Hospital, Sichuan University, Chengdu 610041, China; 2Early Phase Clinical Research Unit, West China Hospital, Sichuan University, Chengdu 610041, China; 3Department of Hepatobiliary Surgery, Chengdu Second People’s Hospital, Chengdu 610011, China; 4Department of Hepatobiliary Surgery, The First Affiliated Hospital of Kunming Medical University, Kunming 650032, China

**Keywords:** no-touch isolation, pancreaticoduodenectomy, pancreatic cancer, long-term outcomes

## Abstract

**Background**: The recurrence and liver metastasis rates are still high in pancreatic head cancer with curative surgical resection. A no-touch isolation principle in pancreaticoduodenectomy (PD) may improve this situation, however, the exact advantages and efficacy of these principles have not been confirmed. **Materials and methods**: Among 370 patients who underwent PD, three centers were selected and classified into two groups: the no-touch PD group (*n* = 70) and the conventional PD group (*n* = 300). Propensity score matching was used to control for selection bias at a ratio of 1:1. The confounding variables were age, sex, body mass index, adjuvant chemotherapy, carbohydrate antigen 19-9, tumor size and tumor differentiation. **Results**: Patients in the no-touch PD group had better overall survival (OS) and disease-free survival (DFS) than those in the conventional PD group (OS: 17 vs. 13 months, *p* = 0.0035, DFS: 15 vs. 12 months, *p* = 0.087), with lower 1- and 2-year disease-related mortality rates (1-year: 32.9% vs. 47%, *p* = 0.032; 2-year: 42.5% vs. 82% *p* = 0.000) and recurrence and liver metastasis rates (1-year: 30.0% vs. 43.3%, *p* = 0.041; 2-year: 34.3% vs. 48.7%, *p* = 0.030). Compared with the matched conventional PD group, the no-touch PD group also had a better OS (17 vs. 12 months, *p* = 0.032). **Conclusions**: Our study showed the no-touch isolation principle may be a better choice to improve long-term survival for pancreatic cancer patients.

## 1. Introduction

Although more than 80% of pancreatic ductal adenocarcinoma (PDAC) patients cannot be cured with surgery [1], surgical resection is still the only curative treatment for patients with PDAC. For tumors in the pancreatic head and neck, pancreaticoduodenectomy (PD) is the standard surgical procedure. The conventional PD procedure requires wide mobilization of the duodenum and pancreatic head, which is called the Kocher maneuver [2]. However, this regular procedure may cause cancer cells to spread into the body via the portal vein, which may increase the risk of early recurrence and metastases [3,4].

Short-term postoperative recurrence and metastases are related to poor prognosis for PDAC patients [5]. Liver metastases (LMs) are the most common and independent factor for poor prognosis [6,7,8,9]. Several studies have indicated that LMs caused by circulating tumor cells (CTCs) that have spread from the primary tumor are the main cause of cancer-related death [10,11,12,13]. No-touch isolation was first proposed in the 1950s to reduce the risk of recurrence in patients with colon cancers, but a recent multicenter randomized phase III trial indicated that no touch was not superior to conventional isolation among patients who were undergoing open primary resection of colon cancers [14].

Since Nakao et al. initially reported the use of the no-touch isolation technique for pancreatic head cancer, several present clinical and basic studies yielded contradictory results of no-touch isolation in PD [15,16,17,18,19,20]. We therefore set this study to evaluate whether no-touch isolation can be superior in terms of postoperative outcome and long-term survival.

## 2. Materials and Methods

### 2.1. Study Design and Participants

This study was a multicenter, retrospective observational study. All patients who underwent no-touch PD or conventional PD at the West China Hospital, the First Affiliated Hospital of Kunming Medical University and the Chengdu Second Peoples Hospital were selected, and their medical data were prospectively collected. This study was in accordance with the ethical principles of the Declaration of Helsinki and the Ethical Guidelines for Medical and Health, and this study was approved by the relevant Ethics Committee. Informed consent was obtained from all patients before enrollment.

All patients who were included in this study had pathologically proven PDAC. Patients with pancreatic solid pseudopapillary neoplasms, intraductal papillary mucinous neoplasms, pancreatic neuroendocrine tumors and cystadenocarcinomas were excluded from the current study.

### 2.2. Surgical Procedure

The concept of the no-touch isolation technique in PD was as follows: (1) The surgeon must not touch the pancreatic head, including the pancreatic cancer, before portal vein (PV) isolation; (2) The gastroduodenal artery (GDA) and inferior pancreaticoduodenal artery (IPDA) are first ligated to intercept the arterial inflow with the early division of the neck of the pancreas; (3) The anterior superior pancreaticoduodenal vein (ASPDV), posterior superior pancreaticoduodenal vein (PSPDV), anterior inferior pancreaticoduodenal vein (AIPDV) and posterior inferior pancreaticoduodenal vein (PIPDV) are then ligated to intercept the venous outflow [2].

In this study, the main procedures of the no-touch PD were summarized as follows: (1) After the exploration of the whole abdominal cavity, the superior mesenteric vein (SMV) was first identified, and the right gastroepiploic vein and Henle’s trunk were dissected. The distal stomach and the pancreatic neck was transected. Then, the GDA was identified along the common hepatic artery (CHA) and ligated; (2) Inspired by the intermediate approach laparoscopic right hemicolectomy, the second step was suspending the transverse colon by clipping the mesocolon to the gallbladder and the liver suspension line to display the inferior part of the duodenum, the uncinate process of pancreas, the main SMV and the first branch of the jejunal vein from the infracolic compartment. The inferior pancreaticoduodenal vein (IPDV), IPDA and superior pancreaticoduodenal vein (SPDV) were ligated successively. We performed a complete clearance of all tissues located approximately 180 degrees to the right of the SMA for a negative margin; (3) After ligating the vessels, we performed a reversed Kocher’s maneuver alongside the inferior vena cava (IVC) to complete the resection of the posterior peritoneum of the duodenum and pancreatic head (Figure 1).

### 2.3. Postoperative Outcomes and Follow-Up

The primary outcome was overall survival (OS), which was defined as the time from surgery to the time of case-related death. The secondary endpoint was disease-free survival (DFS), which was defined as the time from surgery to the first evidence of recurrence. Recurrences were detected according to imaging findings during postoperative follow-up (every 3–6 months for 2 years, then 6–12 months as clinically indicated).

### 2.4. Statistical Analysis

Categorical variables are presented as numbers with percentages and were compared with Person’s chi-square test or Fisher’s exact test. Continuous variables are presented as the means with standard deviations and were compared with the Mann–Whitney U test or Student’s *t*-test. Survival analysis was compared using the Kaplan–Meier method, and differences in survival were analyzed using the log-rank test. A *p*-value < 0.05 was considered significant.

We performed propensity score matching to reduce the potential selection bias between the no-touch PD group and the conventional PD group. The propensity score model was performed using logistic regression analysis, and the variables included in this model were age, sex, body mass index (BMI), adjuvant chemotherapy, carbohydrate antigen 19-9 (CA19-9), tumor size and tumor differentiation. The 1:1 matching was performed using nearest-neighbor matching without replacement with a 0.2 caliper width.

All statistical analyses were performed using the SPSS 26.0 software package (IBM Corp. Released 2013. IBM SPSS Statistics for Windows, Version 22.0.; Armonk, NY, USA: IBM Corp).

## 3. Results

### 3.1. Patient Characteristics

A total of 370 patients with PDAC from August 2014 to July 2021 were enrolled in this study, including 70 patients from January 2019 to July 2021 in the no-touch PD group (48 patients from the First Affiliated Hospital of Kunming Medical University, 18 patients from the West China Hospital of Sichuan University and 4 patients from the Chengdu Second Peoples Hospital) and 300 patients in the conventional PD group (33 patients from the First Affiliated Hospital of Kunming Medical University, 250 patients from the West China Hospital of Sichuan University and 17 patients from the Chengdu Second Peoples Hospital). 

Table 1 shows the baseline characteristics of all patients in both groups. There were no statistically significant differences in age, sex, BMI, adjuvant chemotherapy, preoperative comorbidities, preoperative total bilirubin or tumor size. For preoperative tumor markers, CA19-9 was higher in the conventional PD group, but there were no differences in carcinoembryonic antigen (CEA) and carbohydrate antigen 125 (CA125). After propensity score matching, there were no differences in any baseline characteristics between the two groups.

### 3.2. Postoperative Outcomes

The postoperative results of both groups are summarized in Table 2. The operative time of the no-touch PD group was longer than that of the conventional PD group (430 ± 110 vs. 339 ± 104 min, *p* = 0.000). More patients needed blood transfusion during surgery in the no-touch PD group (38.6% vs. 24%, *p* = 0.013), but there were no differences in blood loss (410 ± 329 vs. 375 ± 367 mL, *p* = 0.140). For postoperative complications, the no-touch PD group had a lower risk of post pancreatectomy hemorrhage (0% vs. 3.7%, *p* = 0.03), and the other complications, such as gastrointestinal bleeding, pancreatic fistula (Grade B/C), biliary fistula, intestinal obstruction and delayed gastric emptying, showed no significant differences between the two groups. Moreover, the postoperative stay of the no-touch PD group was longer than that of the conventional PD group (17 ± 10 vs. 14 ± 11 days, *p* = 0.030). Compared with the matched conventional PD group, there was no difference in both operative information and postoperative complications (Table 2).

Table 3 showed the pathological results of both groups. Patients in the conventional PD group had more lymph node metastases (41.3% vs. 21.4%, *p* = 0.002), more angiolymphatic invasion (26.3% vs. 7.1%, *p* = 0.001) and more perineural invasion (78.0% vs. 55.7%, *p* = 0.000). Besides, patients with poor tumor differentiation were higher in the conventional PD group (18.7% vs. 4.3%), and four patients in the conventional PD group with M1 and stage IV. To eliminate the bias, we added pathological variables into the propensity score model and the matched conventional PD group showed a similar pathological baseline compared with the no-touch PD group.

### 3.3. Long-Term Survival Outcomes

The median OS was significantly different between the no-touch PD group and the conventional PD group (OS: no-touch PD 17 months, conventional PD 13 months, *p* = 0.0035). Although the no-touch PD group had a longer DFS compared with the conventional PD group, the statistical analysis did not show a significant difference (DFS: no-touch PD 15 months, conventional PD 12 months, *p* = 0.197). (Figure 2a,b) The 1- and 2-year disease-related mortality rates of the no-touch PD group and conventional PD group were 32.9% vs. 47% (*p* = 0.032) and 42.5% vs. 82% (*p* = 0.000), respectively. The no-touch PD group also showed a lower recurrence and liver metastasis rate than the conventional PD group (1-year recurrence rate: 30.0% vs. 43.3%, *p* = 0.041; 2-year recurrence rate: 34.3% vs. 48.7%, *p* = 0.030). 

The median OS was also significantly different between the no-touch PD group and the matched conventional PD group (OS: no-touch PD 17 months, matched conventional PD 12 months, *p* = 0.032). However, the DFS of the matched conventional PD group was 11 months, which had no statistical difference compared with the no-touch PD group (*p* = 0.099). (Figure 2c,d) The 1-year and 2-year mortality of the matched conventional PD group were 50.7% and 94.9%, respectively, and the 1-year and 2-year recurrence rate of the matched conventional PD group were 42.0% and 50.7%, respectively.

## 4. Discussion

Based on the results of this study, the overall survival and recurrence-free survival in the no-touch PD group were significantly better than those in the conventional group. However, there were no statistically significant differences between the no-touch PD group and the matched conventional PD group.

For cancers in the ampullary region, pancreaticoduodenectomy is the standard surgical procedure and the only potential curative treatment. However, the postoperative survival rate is still poor due to the high recurrence rate. During conventional PD, the surgeon usually grasps the pancreatic head and duodenum, including the cancers in the pancreatic head, which is called the Kocher maneuver. The Kocher maneuver may squeeze the cancers and increase the risk of spreading the cancer cells into the portal vein, retroperitoneum and peritoneal cavity, which may further lead to liver metastasis, local recurrence and peritoneal dissemination [4].

Surgical manipulation can increase the risk of dissemination of tumor cells into the circulation [21,22,23]. Circulating tumor cells (CTCs) originate from the primary tumor sites and mobilize into the bloodstream and lymphatic system by both intrinsic factors, such as migration and invasion, and extrinsic factors, such as surgical manipulation or biopsy. The detection of CTCs can be used in the follow-up of patients with several cancers (breast, pancreatic, hepatobiliary and colorectal) [24,25,26,27]. Gall TM et al. reported that CTCs were increased in patients who underwent standard PD but not in patients with no-touch isolation PD [20].

To solve this problem and reduce the postoperative recurrence rate, several modifications in PD have been reported. Nakao et al. reported a new technique using an antithrombogenic bypass catheter of the portal vein. The mesenteric venous blood can be bypassed to the systemic circulation or intrahepatic portal vein with this catheter. The dissection of the arteries supplying and veins draining are prior to the manipulation of the pancreatic head and cancers [15]. Kobayashi et al. reported a no-touch isolation technique for PD without removing a portal vein to prevent the manipulated shedding of cancer cells into a portal vein and liver metastasis. They devised this procedure in which the surgeons do not touch the pancreatic head before the isolation of the PV. In this procedure, the surgeons first ligate the GDA and IPDA to block arterial inflow and then ligate the PSPDV, AIPDV and PIPDV to block venous outflow [16].

Recently, several articles have reported the no-touch isolation principle applied in multiple cancers. Kanno K et al. reported that minimally invasive radical hysterectomy (MIRH) with a no-touch isolation technique for stage IA to IB1 cervical cancer was a safe approach with 5-year OS and DFS rates of 97.2% and 96.3%, respectively [28]. Yasukawa M et al. reported that Wedge resection of the tumor site, which is similar to the no-touch isolation technique, could prevent surgical manipulation during lobectomy of non-small cell lung cancer and significantly prolong patients’ overall survival [29]. However, in a multicenter, open-label, randomized, phase III trial that evaluated whether the no-touch technique was superior to the conventional technique in patients with cT3/T4 colon cancer, Takii Y, Mizusawa J et al. failed to confirm the superiority of the no-touch technique [14]. They summarized that the possible explanations were that the no-touch technique is not actually a superior option or that the postoperative survival for colon cancers has improved because of effective standard adjuvant therapy.

We completed this study to verify the efficacy and long-term advantages of no-touch isolation of PD. We hypothesized that the overall survival, disease-free survival and liver relapse-free survival are improved for the no-touch PD group compared with the conventional PD group. Based on our evaluation, patients who underwent no-touch PD had better overall survival and disease-free survival and lower mortality rates and recurrence rates than those who underwent conventional PD.

However, the baseline information of the no-touch PD group and the conventional PD group were significantly different in several variables. There were four patients who underwent conventional PD but were pathologically proved to be M1 with isolated liver metastasis which was not found in the preoperative abdominal imaging. These four patients with worse postoperative OS may lead to bias. To reduce the selection bias between the two groups, we used a propensity score matching model and set a new matched conventional PD group with a ratio of 1:1. The no-touch PD group also had a better prognosis in terms of long-term survival with longer OS compared with the matched conventional PD group. Due to the limitations of a retrospective study, such as the long time-span of the conventional PD group, this conclusion needs further confirmation. Besides, the patients enrolled in this study were from three different centers, the difference of center volume and surgical techniques may lead to the bias of the results. An ongoing randomized control trial registered by our center will further validate this conclusion.

## 5. Conclusions

Our findings demonstrated that no-touch pancreaticoduodenectomy may be superior to the conventional surgical procedure with better overall survival and disease-free survival. A large-volume randomized control trial must be performed to confirm this conclusion.

## Figures and Tables

**Figure 1 jcm-12-00632-f001:**
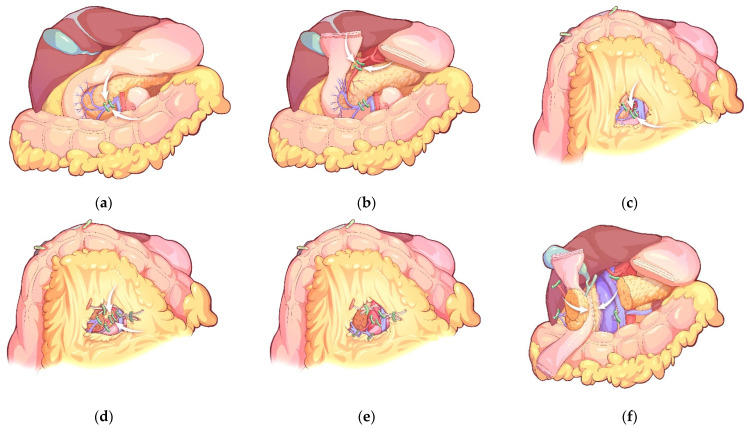
The main surgical procedures of no-touch pancreaticoduodenectomy: (**a**). Ligation of the right gastroepiploic vein and Henle’s trunk; (**b**). Ligation of the gastroduodenal artery; (**c**). Ligation of the inferior pancreaticoduodenal vein; (**d**). Ligation of the inferior pancreaticoduodenal artery; (**e**). Complete clearance of all tissues located around 180 degrees to the right of the superior mesenteric artery; (**f**). Complete the resection of the posterior peritoneum of the duodenum and pancreatic head by a reversed Kocher’s maneuver.

**Figure 2 jcm-12-00632-f002:**
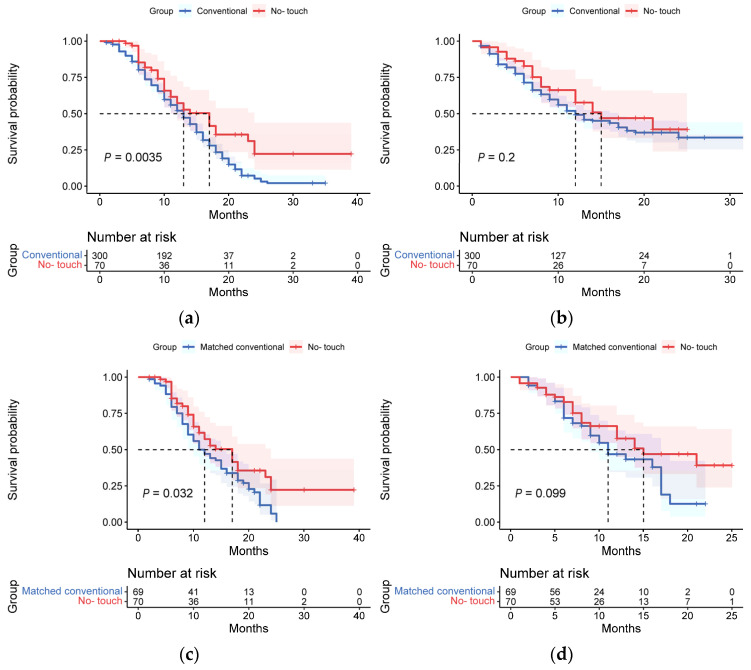
Long-term survival outcome between no-touch PD group and conventional PD group: (**a**) Overall survival between no-touch PD group and conventional PD group; (**b**) Disease-free survival between no-touch PD group and conventional PD group; (**c**) Overall survival between no-touch PD group and matched conventional PD group; (**d**) Disease-free survival between PD group and matched conventional PD group.

**Table 1 jcm-12-00632-t001:** Patient Characteristics.

	No-Touch PD Group (*n =* 70)	Conventional PD Group (*n =* 300)	Matched Conventional PD Group (*n =* 70)	*p*-Value ^a^	*p*-Value ^b^
Age (years), mean ± SD	59.93 ± 9.35	60.24 ± 10.70	61.22 ± 9.37	0.829	0.418
Sex				0.560	0.324
Male	41 (58.6%)	187 (62.3%)	46 (66.7%)		
Female	29 (41.4%)	113 (37.7%)	23 (23.3%)		
BMI (kg/m^2^)	22.11 ± 3.80	21.88 ± 3.36	21.90 ± 3.52	0.583	0.732
Adjuvant therapy	23 (32.9%)	117 (39.0%)	17 (24.6%)	0.340	0.285
Total bilirubin (mol/L)				0.580	0.571
<28	22 (31.4%)	111 (37.0%)	25 (35.7%)		
28–34	4 (5.7%)	5 (1.7%)	1 (1.4%)		
34–171	22 (31.4%)	96 (32.0%)	21 (30.4%)		
>171	22 (31.4%)	88 (29.3%)	22 (31.9%)		
Comorbidities, *n* (%)					
Hypertension	15 (21.4%)	59 (19.7%)	13 (18.8%)	0.426	0.704
Diabetes mellitus	7 (10%)	43 (14.3%)	10 (14.3%)	0.227	0.438
Tumor size (cm)	3.13 ± 1.49	3.25 ± 0.98	3.25 ± 1.10	0.067	0.609
Tumor marker					
CA19-9	268.89 ± 332.61	389.53 ± 378.54	260.27 ± 297.84	0.009	0.873
CA125	27.60 ± 24.56	29.02 ± 31.68	31.03 ± 30.42	0.552	0.442
CEA	5.38 ± 5.58	7.15 ± 18.89	5.26 ± 6.77	0.992	0.903

^a^ No-touch PD group vs. Conventional PD group; ^b^ No-touch PD group vs. Matched conventional PD group.

**Table 2 jcm-12-00632-t002:** Operative results.

	No-Touch PD Group (*n =* 70)	Conventional PD Group (*n =* 300)	Matched Conventional PD Group (*n =* 70)	*p*-Value ^a^	*p*-Value ^b^
Operative information					
Laparoscopic PD/Open PD	26/44	35/265	24/45	0.000	0.772
Time (min)	432 ± 110	339 ± 104	410 ± 88	0.000	0.178
Blood loss (mL)	410 ± 329	375 ± 367	360 ± 353	0.140	0.390
Blood transfusion, *n* (%)	27 (38.6%)	72 (24.0%)	23 (33.3%)	0.013	0.520
PV/SMV resection, *n* (%)	7 (10.0%)	47 (15.7%)	5 (7.2%)	0.227	0.563
Postoperative complications					
Post-pancreatectomy hemorrhage	0 (0.0%)	11 (3.7%)	2 (2.9%)	0.030	0.151
Gastrointestinal bleeding	1 (1.4%)	9 (3.0%)	4 (5.8%)	0.432	0.167
Pancreatic fistula	9 (12.9%)	45 (15%)	14 (20.3%)	0.647	0.238
Biliary fistula	2 (2.9%)	11 (3.7%)	1 (1.4%)	0.734	0.568
Intestinal obstruction	2 (2.9%)	4 (1.3%)	2 (2.9%)	0.398	0.988
Delayed gastric emptying	6 (8.6%)	33 (11%)	7 (10.1%)	0.551	0.750
Postoperative hospital stay (days)	17 ± 10	14 ± 11	16 ± 9	0.030	0.496
Reoperation	3 (4.3%)	10 (3.3%)	2 (2.9%)	0.704	0.661

^a^ No-touch PD group vs. Conventional PD group; ^b^ No-touch PD group vs. Matched conventional PD group.

**Table 3 jcm-12-00632-t003:** Pathological Results.

	No-Touch PD Group (*n =* 70)	Conventional PD Group (*n =* 300)	Matched Conventional PD Group (*n =* 70)	*p*-Value ^a^	*p*-Value ^b^
Pathological results					
Tumor size, cm	3.13 ± 1.49	3.25 ± 0.98	3.25 ± 1.10	0.067	0.609
Tumor differentiation				0.000	0.138
Well	12 (17.1%)	5 (1.7%)	5 (7.2%)		
Moderately	55 (78.6)	239 (79.7%)	58 (84.1%)		
Poorly	3 (4.3%)	56 (18.7%)	6 (8.7%)		
Lymph node metastases	15 (21.4%)	124 (41.3%)	19 (27.5%)	0.002	0.402
Angiolymphatic invasion	5 (7.1%)	79 (26.3%)	8 (11.6%)	0.001	0.367
Perineural invasion	39 (55.7%)	234 (78.0%)	46 (66.7%)	0.000	0.185
AJCC 8th Stage					
T stage (T1/T2/T3/T4)	15/40/15/0	34/219/45/2	10/47/12/0	0.040	0.389
N stage (N0/N1/N2)	55/14/1	176/99/25	50/15/4	0.050	0.356
M stage (M0/M1)	70/0	296/4	69/0	0.331	
Stage (I/II/III/IV)	35/34/1/0	149/120/27/4	41/24/4/0	0.102	0.136

^a^ No-touch PD group vs. Conventional PD group; ^b^ No-touch PD group vs. Matched conventional PD group.

## Data Availability

Not applicable.

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
