# Peer review of "Long Term Outcomes of No-Touch Isolation Principles Applied in Pancreaticoduodenectomy for Treatment of Pancreatic Adenocarcinoma: A Multicenter Retrospective Study with Propensity Score Matching"

_jcm, 2023, doi:10.3390/jcm12020632_

Round 1
Reviewer 1 Report
In this manuscript, the authors examined the benefits of non-touch pancreatoduodenectomy (PD) for pancreatic cancer by comparison with conventional PD, mainly focused on survival analysis. They showed that the patients in the no-touch PD group had better overall and disease free survival than the conventional PD group. The propensity score matched analysis could not confirm the significant differences in the overall and disease free survival. They concluded no-touch PD might be a better choice to improve long-term survival.
The comparison between no-touch and conventional PD is fascinating for pancreatic surgeons. However, there are several weak points.
They could not confirm the survival benefit of no-touch PD in the propensity score matched study. However, they concluded the no-touch PD might be a better choice. I could not understand this conclusion. The comparison, including all patients, showed the differences in patients’ backgrounds. In fact, Table 1 showed CA19-9 level was significantly higher in the conventional PD group than in the no-touch PD group. Tumor size also tended to be bigger in the conventional PD group than in the no-touch PD group. In Table 3, the conventional PD group also included more node-positive cases and angiolymphatic invasion than the no-touch PD group. These data showed that the conventional PD group might have more advanced cancer than the no-touch PD group, resulting in a poor prognosis. They did not reveal how they selected the surgical techniques. Such background differences, but not surgical technique, might lead to poor prognosis.
In propensity matched analysis, overall and disease free survival was very similar in both groups (Fig 2 c and d). Taking these results, I think their conclusion will mislead the readers. They should change the conclusion.
Author Response
Response to Reviewer 1 Comments
Point 1: They could not confirm the survival benefit of no-touch PD in the propensity score matched study. However, they concluded the no-touch PD might be a better choice. I could not understand this conclusion. The comparison, including all patients, showed the differences in patients’ backgrounds. In fact, Table 1 showed CA19-9 level was significantly higher in the conventional PD group than in the no-touch PD group. Tumor size also tended to be bigger in the conventional PD group than in the no-touch PD group. In Table 3, the conventional PD group also included more node-positive cases and angiolymphatic invasion than the no-touch PD group. These data showed that the conventional PD group might have more advanced cancer than the no-touch PD group, resulting in a poor prognosis. They did not reveal how they selected the surgical techniques. Such background differences, but not surgical technique, might lead to poor prognosis.
In propensity matched analysis, overall and disease free survival was very similar in both groups (Fig 2 c and d). Taking these results, I think their conclusion will mislead the readers. They should change the conclusion.
Response 1:
Dear Reviewer:
Thanks for your work. In the previous manuscrpit we did not show the OS and DFS of the matched conventioal PD group. We have corrected this sectoin in the “Long-term Survival Outcomes” paragraph. The OS of the matched conventional PD group was 13 months too, but the stastistical analysis between the no-touch PD group (OS: 21 months) and the matched conventional PD group (OS: 13 months) showed the P value was 0.23.
According to another reviewer’s comments, we perforemd a new propensity score matching model with added pathological variables. The new matched conventiaonl PD group showed a significantly lower OS compared with the no-touch group (12 vs. 17 months, P=0.032). All changes in the manuscript were marked in red.
Sincerely yours,
Nengwen Ke
West China Hospital, Sichuan University

Reviewer 2 Report
Dear Authors,
The manuscript is interesting and it describes a topic that is not widely discussed in the literature. The description of the no-touch technique is well-written and illustrated, which is a merit of the article. However, I found the design of the study flawed and the discussion does not touch on the serious weaknesses of the study:
- The conventional group had significantly more poorly differentiated tumors, and significantly more tumors with angiolymphatic and perineural invasion. These differences were significant even in the matched groups and they are crucial for the overall survival and disease -free survival. The authors give the histopathological data only in the table and do not discuss it further. Because of these differences between the groups, the study does not justify the conclusion that no-touch technique may be a better approach. The studied groups need to be matched also according to the histopathological features of the tumors.
- The authors describe thoroughly the no-touch method. Were all no-touch surgeries performed in one center and the conventional PDs in other centers, or did all centers apply the same no-touch technique? How many patients were operated in each center – are they all large – volume centers? This should be described in the Methods section.
- The study included patients operated on during 7 years. The operative technique could have improved during this time, this should be shortly discussed.
- Why did 4 patients with M1 undergo PD, which is not indicated for metastatic pancreatic cancer?
- Conventional PD had significantly shorter hospital stay, shorter operative time and significantly less blood loss and less blood transfusions. Blood transfusions have a significant impact on cancer recurrence, this should be discussed.
- “A large-volume randomized control trial was performed to confirm this conclusion.” – probably the authors meant that such trial “must be performed”.
Author Response
Response to Reviewer 2 Comments
Point 1: The conventional group had significantly more poorly differentiated tumors, and significantly more tumors with angiolymphatic and perineural invasion. These differences were significant even in the matched groups and they are crucial for the overall survival and disease -free survival. The authors give the histopathological data only in the table and do not discuss it further. Because of these differences between the groups, the study does not justify the conclusion that no-touch technique may be a better approach. The studied groups need to be matched also according to the histopathological features of the tumors.
Response 1: We perforemd a new propensity score matching model with added pathological variables.
Point 2: The authors describe thoroughly the no-touch method. Were all no-touch surgeries performed in one center and the conventional PDs in other centers, or did all centers apply the same no-touch technique? How many patients were operated in each center – are they all large – volume centers? This should be described in the Methods section.
Response 2: We described and modified these information in the methods section.
Point 3: The study included patients operated on during 7 years. The operative technique could have improved during this time, this should be shortly discussed.
Why did 4 patients with M1 undergo PD, which is not indicated for metastatic pancreatic cancer?
Response 3: We modified our disccusion section to describe this point. The 4 patients with M1 were isolated liver metastases which were found intraoperatively.
Point 4: Conventional PD had significantly shorter hospital stay, shorter operative time and significantly less blood loss and less blood transfusions. Blood transfusions have a significant impact on cancer recurrence, this should be discussed.
Response 4: We perforemd a new propensity score matching model with added pathological variables and the new matched conventional PD group had the similar pathological baseline compared with the no-touch PD group
Dear Reviewer:
Thanks for your work. All changes in the manuscript were marked in red.
Sincerely yours,
Nengwen Ke
West China Hospital, Sichuan University

Round 2
Reviewer 2 Report
Dear Authors, thank you for the corrections. I still have some minor comments:
The Methods section lacks a few important points. Discussion is still very short, repeats the results without discussing any problematic points and referring them to the literature.
- The authors only included information about the numbers of no-touch PDs in the hospitals taking part in the study. One hospital had only 4 no-touch PDs during the 7 years period included in the study. Were there such differences in conventional PDs as well? The numbers of patients operated on in each center in both groups should be disclosed in the Methods section and discussed in the Discussion paragraph as a potential source of bias (low-volume centers having generally worse outcomes than large-volume centers is an important issue in a study assessing a new surgical technique).
- Also, the number of no-touch PDs (70) is higher than the number of no-touch PD patients from the three centers (48+12+4), there must be some error.
- In the discussion it is mentioned that the no-touch technique has been used for a considerably shorter time than the conventional technique (“the long time span of the conventional PD group”). The Materials and Methods section should mention exactly, since when the no-touch technique is practiced in the hospitals involved.
- The 4 patients with M1 should also be mentioned in the discussion, because the continuation of PD after the intraoperative finding of liver metastases is still doubtful and these patients probably had significantly worse outcomes. Referral to literature justifying these resections would be commendable.
- The added paragraph (lines 150-154) requires some English correction.
- Figure 2:
(b) Disease-free survival between PD group and conventional PD group;– please correct the legend (no-touch PD and conventional PD)
Author Response
Point 1: The Methods section lacks a few important points. Discussion is still very short, repeats the results without discussing any problematic points and referring them to the literature.
- The authors only included information about the numbers of no-touch PDs in the hospitals taking part in the study. One hospital had only 4 no-touch PDs during the 7 years period included in the study. Were there such differences in conventional PDs as well? The numbers of patients operated on in each center in both groups should be disclosed in the Methods section and discussed in the Discussion paragraph as a potential source of bias (low-volume centers having generally worse outcomes than large-volume centers is an important issue in a study assessing a new surgical technique).
- Also, the number of no-touch PDs (70) is higher than the number of no-touch PD patients from the three centers (48+12+4), there must be some error.
- In the discussion it is mentioned that the no-touch technique has been used for a considerably shorter time than the conventional technique (“the long time span of the conventional PD group”). The Materials and Methods section should mention exactly, since when the no-touch technique is practiced in the hospitals involved.
- The 4 patients with M1 should also be mentioned in the discussion, because the continuation of PD after the intraoperative finding of liver metastases is still doubtful and these patients probably had significantly worse outcomes. Referral to literature justifying these resections would be commendable.
- The added paragraph (lines 150-154) requires some English correction.
- Figure 2:
(b) Disease-free survival between PD group and conventional PD group;– please correct the legend (no-touch PD and conventional PD)
Response 1:
Dear reviewer:
We have revised the article according to your comments, and the modified places are marked in red. Thanks for your hard word. Wish you all the best in the coming new year.
Sincerely yours.
Nengwen Ke
West China Hospital, Sichuan University
